A systematic review of effects of daytime napping strategies on sports performance in physically active individuals with and without partial-sleep deprivation

Sirohi Priya 1
Khan Moazzam Hussain 1 mkhan47@jmi.ac.in
Sharma Saurabh 1
http://orcid.org/0000-0002-7868-5113 Nuhmani Shibili 2
Al Muslem Wafa Hashem 2
Abualait Turki 2
1 Centre for Physiotherapy and Rehabilitation Sciences, Jamia Millia Islamia , New Delhi , India
2 Department of Physical Therapy, Imam Abdulrahman Bin Faisal University , Dammam, Eastern Province , Saudi Arabia
Badicu Georgian
Electronic publication date: 2022 Dec 1
Publication date: 2022
Volume: 10
Electronic Location ID: e14460
Received 2022 May 19; Accepted 2022 Nov 3
Copyright: © 2022 Sirohi et al.
Copyright year: 2022
Copyright holder: Sirohi et al.
License: This is an open access article distributed under the terms of the Creative Commons Attribution License, which permits unrestricted use, distribution, reproduction and adaptation in any medium and for any purpose provided that it is properly attributed. For attribution, the original author(s), title, publication source (PeerJ) and either DOI or URL of the article must be cited.
License URL: https://creativecommons.org/licenses/by/4.0/

Keywords: Mid-day sleep, Nap, Sleep restriction athlete, Recovery, Sports performance, Cognitive performance

Funding: The authors received no funding for this work.

==============================
Background

Sleep is the body’s natural recovery process, restoring routine metabolic and regulatory functions. Various sleep interventions have been developed to facilitate recovery, and athletic performance, and daytime napping are among them. However, due to inconsistencies in studies, it remains unclear whether daytime napping affects sports performance. This article aims to review the effects of daytime napping on various variables of sports performance in physically active individuals with and without partial-sleep deprivation.

Methods

A systematic search in three clinical databases, namely Cochrane Central Register of Controlled Trials (CENTRAL), PubMed, and Web of Science, was conducted. To be included in the current review, the study should be a randomized controlled trial that evaluated the influence of daytime napping on one or more components of sports performance in healthy adults, 18 years old or older.

Results

In the accessible data available until December 2021, 1,094 records were found, of which 12 relevant randomized controlled trials were selected for qualitative synthesis. The majority of studies reported favourable effects of daytime napping on sports performance. However, only one study reported no significant impact, possibly due to a different methodological approach and a shorter nap duration.

Conclusion

Napping strategies optimize sports performance in physically active, athletic populations, benefitting partially sleep-deprived and well-slept individuals, with longer nap durations (~90 min) having more significant advantages. Daytime naps can be considered as cost-efficient, self-administered methods promoting recovery of body functions.

Introduction

Sleep, the most basic of biological activities in humans, is defined as a physiological process in which the body’s metabolic, among other regulatory mechanisms, slows down for some time, allowing recovery and preparing the body for upcoming metabolic and regulatory processes (Aldabal & Bahammam, 2011). The sleep-wake continuum is regulated by intrinsic biological clocks present in suprachiasmatic nuclei (SCN) of the hypothalamus (Van Dongen & Dinges, 2000). SCN also synchronizes the circadian fluctuations of physiological functions, including alertness, cognitive abilities, body temperature, blood pressure, hormones, and physical performance (Davenne, 2009).

Sleep can be differentiated into rapid eye movement (REM) sleep and non-REM sleep. The non-REM, which is also referred as slow-wave, is a component of sleep that offers energy conservation, decreases stress and anxiety, and thus, aids in good recovery (Mulrine et al., 2012). Throughout a night’s sleep, multiple sets of REM, non-REM and awake states occur at different points of time, which is critical for the functioning of cortical centres, among other body functions, and promote recovery (Bonnet, Berry & Arand, 1991). Furthermore, outcomes of athletic performance critically depend on the coping abilities to counteract physiological and psychological stressors (Bishop, 2008). Thus, sleep has been identified as a crucial component in both physiological and psychological terms in the athletic population (Dickinson & Hanrahan, 2009) and is considered the single best recovery method for athletes (Halson, 2008).

Total sleep deprivation is the state of wakefulness for more than 24 h, leading to extreme sleep loss. Partial sleep deprivation is the decrease in total sleep time i.e., either waking up earlier than normal or falling asleep later (Alhola & Polo-Kantola, 2007). Sleep deprivation can significantly affect sports performance as it is potentially associated with reduced production of aerobic and anaerobic power (Reilly & Edwards, 2007; Guezennec et al., 1994). Prolonged sleep deprivation (~36 h) is linked to an increase in sympathetic and decrease in parasympathetic cardiovascular modulation, and baroreflex sensitivity during sitting and vigilance testing in healthy adults (Zhong et al., 2005). As overtraining is associated with autonomic imbalance (Achten & Jeukendrup, 2003). These disturbances could lead to the development of over-reaching or over-training (Hynynen et al., 2006).

Factors influencing sleep in sportspersons include the timings of the competition (Fullagar et al., 2016), post-training and competition muscle pain and tension (Halson, 2014), raised core temperature (Oda & Shirakawa, 2014; Chennaoui et al., 2015), sound and light disturbances (Romyn et al., 2016), psychological stress and other social requirements, which can misbalance the thermo-physiological cascade of sleep initiation (Nédélec et al., 2015; Kräuchi, 2007). Therefore, those athletes who are routinely participating in extensive training and competitions throughout the year have an increased prevalence of sleep inadequacy. As reported by the researchers, global sleep quality indicates the sleep disturbances experienced by 50–78% of elite sportspersons, with 22–26% of athletes having highly disrupted sleep (Gupta, Morgan & Gilchrist, 2017; Samuels, 2008; Swinbourne et al., 2016).

The researchers have developed advanced sleep interventions to improve poor sleep patterns and optimize performance and recovery measures. These sleep interventions are broadly divided into post-exercise recovery methods, napping strategies, and sleep hygiene (Bonnar et al., 2018). The practice of sleep hygiene targets sleep-related behaviours to improve good sleep at night (Harada et al., 2016). As supported by literature, sleep is also affected by post-exercise recovery methods (Schaal et al., 2015). Lastly, napping strategies focus on improving sleep acquired through brief targeted naps or total sleep durations (Mah et al., 2011).

Napping can be defined as a period that is less than 50% of the total nocturnal sleep duration (Dinges et al., 1987). Therefore, napping is considered to be a period of revitalization. The tendency to sleep in response to the post-lunch period of sleepiness suggests human cognitive performance follows a circadian rhythm accompanied by performance dips during the afternoon with peaks in the early evening (Schmidt et al., 2007). This period of sleepiness occurs between 13:00 to 16:00 h with a slight decrease in core temperature, encouraging sleep propensity (Van Dongen & Dinges, 2000). Sleep loss, fatigue, and stress can increase the measure of sleepiness due to post-lunch dip (Winget, DeRoshia & Holley, 1985) and thus, impact athletic performance either during training or competing in the afternoon (Nédélec et al., 2015).

Multiple research studies have suggested that napping approaches have contributed to the improvement of the performance of athletes and sportspersons and improved the sports-related parameters, i.e., improved jump velocity, endurance performance, karate specific test, counter-movement as well as squat jumps, 5-min shuttle run, etc. (Blanchfield et al., 2018; Daaloul, Souissi & Davenne, 2019; Boukhris et al., 2020). However, one study demonstrated an insignificant effect of a 20-min nap opportunity on power output during the Wingate test, i.e., after normal sleep or the 5-h phase of expanded sleep conditions in the athletic population (Petit et al., 2014).

This article aims to review and study the impacts of daytime napping on various variables of sports performance in physically active individuals with and without partial-sleep deprivation. The information from this systematic review of literature will impart crucial insights to the domains of sports science and sleep medicine through an attempt to put forward the key idea of sleep strategies, namely, daytime napping, as an effective recovery method in physically active population with and without partial-sleep deprivation and its influence on sports-related outcomes.

Methodology

The statement and guidelines of the Preferred Reporting Items for Systematic Reviews and Meta-Analyses (PRISMA) are referred for the present systematic review of the literature (Moher et al., 2009).

Material sources and search

A systematic search is conducted to retrieve data, available until December 2021, from three databases: Cochrane Central Register of Controlled Trials (CENTRAL), PubMed and Web of Science. The keywords “daytime napping”, “athlete”, “performance”, “recovery”, and “nap opportunity” with no additional filters were used in the search.

Eligibility criteria

Peer-reviewed articles published in English were selected for inclusion in the study. In addition, randomized controlled trials which evaluated the influence of daytime napping on one or more components of sports performance in healthy adults, 18 years old or older, were included. However, those articles, which pertain to data on (1) infants, children, adolescents, elderly population, shift-workers, non-healthy adults, and animal subjects; (2) effect of jet lag, stimulants (e.g., caffeine), and pharmacological interventions, were excluded. In addition, observational studies, non-randomized clinical trials, review articles, case series, letters to the editor, dissertation/thesis reports, meeting abstracts and conference proceedings were also excluded.

Selection of studies

The authors, PS and MHK, screened the title and abstract of individual retrieved records after removing duplicates. Full-text papers were then independently screened by both the authors based on the pre-designed acceptability criteria to extract eligible articles to be included in the present study. Finally, the authors collected data from the selected studies into an MS-Word data collection table designed to record information on each study. Disagreements between the authors were resolved through a mutual consensus.

Data collection

Data about authors and year of study, participants (mean age, sex, and status of habitual napping), study design, sleep deprivation (sleep-deprived hours and timing of sleep), intervention (duration and timing of nap, control or comparison group/condition), test timing, outcome measures, and results (significant p-value and major findings) were extracted.

Quality assessment

The quality assessment of selected studies was conducted through a broad set of items of an 11-point PEDro scale designed to assess the methodological and scientific quality of randomized clinical studies (Verhagen et al., 1998). Two authors independently evaluated the quality of the studies. Discrepancies between the authors’ decision scores were discussed and resolved to their mutual satisfaction.

All the studies met the first criterion specified in the eligibility criteria, so this was not included in the scoring. Instead, each of the studies was given a score for meeting the remaining ten criteria. If the criterion was completed, the score was 1; if not, the score was 0. The quality of studies based on the total scoring was categorized as excellent (score > 8), good (score of 6–8), fair (score of 4 or 5) and, poor (score < 4) (Hariohm, Prakash & Saravankumar, 2015).

Results

A total of 1,094 records were identified, out of which 479 duplicate records were removed. The remaining titles and abstracts were screened, out of which 46 full-text articles were selected following the inclusion/exclusion criteria. Three records were added after a manual search of reference lists of potential full-text articles. Twelve studies were finally included for qualitative synthesis.

Figure 1 depicts the flow chart and outcomes of the literature examination. Tables 1 and 2 summarized the characteristics of trials included for this review.

Figure 1 PRISMA flow diagram.

Table 1 Summary of selected studies (nap after partial-sleep deprived condition).

Reference	Population	Habitual nappers	Study design	Sleep deprivation (Sleep-deprived duration and timing of sleep)	Intervention
(Duration and timing of Nap)	Control/ comparison group or condition	Test timing	Outcome measures	Significance (p-value)	Major findings	
Waterhouse et al. (2007)	10 healthy males (mean age: 23.3 ± 3.4 y)	NS	Randomized controlled trial (crossover design)	4 h; 22:30– 23:30 h to 02:30–03:30 h	30 min; 13:00–13:30 h (Nap)	No nap	14:00 h	Handgrip strength
2-m Sprint time
20-m Sprint time	p = 0.023
p = 0.80 (ns)
p = 0.063
p = 0.031
p = 0.013	Left vs. right hand
Nap vs. no nap
Between tests 1–3
Fall in mean time from
tests 1–3	
Hammouda et al. (2018)	9 highly trained male judokas (mean age: 18.51 ± 0.93 y)	No	Randomized controlled trial	22:00–02:30 h	20 min; (14:10–14:30 h) (N20)
90 min; (13:00–14:30 h) (N90)	No nap	15:00 h	RAST
Pmax
Pmin
Pmean	p < 0.001
p < 0.001
p = 0.007
p = 0.028
p < 0.001
p = 0.018
p = 0.008
p < 0.001
p = 0.001	N20> no nap
N90> no nap
N90> N20
N20> no nap
N90> no nap
N90> N20
N20> no nap
N90> no nap
N90> N20	
											
Daaloul, Souissi & Davenne (2019)	13 male karate athletes (mean age: 23 ± 2 y)	Yes	Randomized controlled trial	4 h; 23:00– 03:00 h (PSD)	30 min; 13:00–13:30 h (Nap)	13:00–13:30 h (No nap)	14:00– 17:00 h	SJ
CMJ
Time to exhaustion during KST	p > 0.05 (ns)
p < 0.01
p > 0.05 (ns)
p < 0.05
p < 0.001
p > 0.05 (ns)	No significant effect of nap
Nap > Non-nap (post-KST fatigue)
No significant effect of nap
Nap > Non-nap (post-KST fatigue)
Nap after PSD
Nap after RN	
Brotherton et al. (2019)	15 male participants (mean age: 22.7 ± 2.5 y)	No	Randomized controlled trial	3 h; 03:30– 06:30 h (SD)	1 h; 13:00– 14:00 h (SDN)	Normal night sleep with no nap (N) and
SD with no nap	17:00 h	Grip strength
Bench press-
AP
AF
PV
tPV & D
Leg press-
AP
AF, PV, tPV & D	p = 0.041
p = 0.002
p = 0.53 (ns)
p < 0.0005
p = 0.002
p = 0.007
p < 0.0005
p < 0.005
p > 0.05 (ns)
p = 0.002
p = 0.031
p = 0.65 (ns)
ns	N > SD
SDN > SD
Between N & SDN
N > SD
SDN > SD
N > SD
N > SD
SDN > SD
Between N & SD
N > SD
SDN > SD
Between N & SD
Non-significant differences in all three conditions	
Romdhani et al. (2020)	9 highly trained male judokas (mean age: 18.78 ± 1.09 y)	No	Randomized controlled trial	4 h; 22:00– 02:30 h (PSD)	20 min; 14:10–14:30 h (N20)
90 min; 13:00–14:30 h (N90)	Normal sleep night (NSN),
PSD and
No nap after PSD	15:00 h	RAST-
Pmax
Pmin
Pmean
FI	p < 0.001
p =0.021
p < 0.001
p < 0.001
p < 0.001
p < 0.001
p < 0.001
p = 0.006	PSD < NSN
N20 > PSD
N90 > PSD
PSD < NSN
N90 > PSD
PSD < NSN
N90 > PSD
N90 < no nap after PSD	
Ajjimaporn, Ramyarangsi & Siripornpanich (2020)	11 trained male collegiate soccer players (mean age: 20 ± 1 y)	No	Randomized controlled trial (crossover design)	3 h; 22:30– 2:00 h (SD)	20 min; 13:00–13:20 h (NaP)	Normal sleep condition (22:30–7:30 h) (CN) and
Sleep deprived condition (22:30–2:00 h) (SD)	16:00 h	RAST-
Pmax
Pmin
Pmean
FI
Leg muscle strength	p = 0.01
p = 0.003
p = 0.04
p = 0.0004
p = 0.0005
p = 0.03
p = 0.04
p = 0.002
p = 0.03
p = 0.02	SD < CN
NaP < CN
SD < CN
NaP > SD
SD < CN
NaP > SD
SD < CN
NaP < CN
SD < CN
Between Nap & SD	
Note:

y, years; NS, Not specified; h, hour; min, minutes; ns, Non-significant; ↑, increase; ↓, decrease; ↔, no effect; RAST, Running-based Anaerobic Sprint Test; Pmax, Highest power; Pmin, Lowest power; Pmean, Sum of all six powers/6; PSD/SD, Partial sleep deprivation; SJ, Squat jump; CMJ, Counter movement jump; KST, Karate specific test; RN, Reference night; SD, Sleep deprivation; SDN, Nap after partial sleep deprivation; AP, Average power; AF, Average force; PV, Peak velocity; tPV, Time-to-peak velocity; D, Distance; FI, Fatigue index; CN, Normal sleep; PTN1, Normal sleep condition with post-lunch rest; PTN2, Normal sleep condition with post-lunch nap; PTPAN1, 5-h phase advance condition with post-lunch rest; PTPAN2, 5-h phase advance condition with post-lunch nap; CON, control group; TTE, Time to exhaustion; BD, Best distance; TD, Total distance; FI, Fatigue index; HD, Highest distance; MVIC, Maximum Voluntary Isometric Contraction.

Table 2 Summary of selected studies (nap after no sleep-deprived condition).

Reference	Population	Habitual nappers	Study design	Intervention	Control/ comparison group or condition	Test timing	Outcome measures	Significance (p-value)	Major findings	
Petit et al. (2014)	16 highly trained male subjects (mean age: 22.2 ± 1.7 y)	No	Randomized controlled trial	60 min; 13:00–14:00 h (PTN2, C)
60 min; 08:00–09:00 h (PTPAN2, D)	60 min rest (PTN1, A)
60 min rest
(PTPAN1, B)	A & C:
Trial 1- 15:30 h
Trial 2-
17:30 h
B & D:
Trial 1-10:30 h
Trial 2-
12:30 h	Wingate test-
Peak power
Mean power
FI	ns
ns
ns	Non-significant effect of nap and phase-advance conditions.	
Blanchfield et al. (2018)	11 trained male runners (mean age: 35 ± 12 y)	NS	Randomized controlled trial (crossover design)	20 ± 10 min; between 14:00–16:50 h (NAP)	CON	Morning exercise session- 08.48 ± 01:09 h
Evening exercise session-
17:03 ± 00:50 h	Treadmill running-
Running TTE
Night-time sleep & TTE	p = 0.83 (ns)
p < 0.01
p = 0.001	Between NAP & CON
Improved post-NAP TTE in subjects with <7 h of night-time sleep
NAP > CON (predicted changes in TTE in subjectsz with <7 h of night-time sleep)	
Boukhris et al. (2019)	17 physically active men (mean age: 21.3 ± 3.4 y)	NS	Randomized controlled Trial	25 min;14:00– 14:25 h (N25)
35min; 14:00–14:35 h (N35)
45 min; 14:00–14:45 h (N45)	No-nap control (N0)	17:00 h	5-m shuttle run test-
BD
TD
FI	p = 0.03
p < 0.0005
p = 0.46 (ns)
p = 0.001
p = 0.01
p = 0.009
p < 0.000
p = 0.001
p < 0.0005
p = 0.18 (ns)	N25 > N0
N45 > N0
Between N35 & N0
N45 > N35
N25 > N0
N35 > N0
N45 > N0
N45 > N25
N45 > N35	
Abdessalem et al. (2019)	18 physically active men (mean age: 21 ± 3 y)	NS	Randomized controlled Trial	25 min Nap Opportunity at 13:00 h
14:00 h
15:00 h	no-nap opportunity	17:00 h	5-m shuttle run test-
TD
HD	p < 0.05
p < 0.05
p < 0.01
p < 0.05
p < 0.01	14:00 h > no-nap
14: 00 h > 13:00 h
14:00 h > no-nap
15:00 h > no-nap
14:00, 15:00 h > 13:00 h	
Hsouna et al. (2019)	20 physically active males (mean age: 21.1 ± 3.6 y)	NS	Randomized controlled trial	25 min; 14:00–14:25 h (N25)
35min; 14:00–14:35 h (N35)
45 min; 14:00–14:45 h (N45)	No-nap opportunity (N0)	17:00 h	5-jump test-
Mean stride	p < 0.01
p < 0.01
ns	N35 > N0
N45 > N0
Between N25 & N0	
Boukhris et al. (2020)	14 amateur team sports players (mean age: 20.3 ± 3.0 y)	NS	Randomized controlled trial (crossover repeated-measures design)	40 min; 14:00–14:40 h (N40)
90 min; 14:00–15:30 h (N90)	No-nap (N0)	17:00 h	MVIC
5-m Shuttle run test-
HD
TD
FI	p < 0.0005
p < 0.0005
p < 0.0005
p < 0.0005
p < 0.0005
p < 0.0005
p < 0.0005
p = 0.04
p = 0.001	N40 > N0
N90 > N0
N90 > N40
N40 > N0
N90 > N0
N40 > N0
N90 > N0
N90 > N40
N90 > N0	
Note:

y, years; NS, Not specified; h, hour; min, minutes; ns, Non- significant; ↑, increase; ↓, decrease; ↔, no effect; RAST, Running-based Anaerobic Sprint Test; Pmax, Highest power; Pmin, Lowest power; Pmean, Sum of all six powers/6; PSD/SD, Partial sleep deprivation; SJ, Squat jump; CMJ, Counter movement jump; KST, Karate specific test; RN, Reference night; SD, Sleep deprivation; SDN, Nap after partial sleep deprivation; AP, Average power; AF, Average force; PV, Peak velocity; tPV, Time-to-peak velocity; D, Distance; FI, Fatigue index; CN, Normal sleep; PTN1, Normal sleep condition with post-lunch rest; PTN2, Normal sleep condition with post-lunch nap; PTPAN1, 5-h phase advance condition with post-lunch rest; PTPAN2, 5-h phase advance condition with post-lunch nap; CON, control group; TTE, Time to exhaustion; BD, Best distance; TD, Total distance; FI, Fatigue index; HD, Highest distance; MVIC, Maximum Voluntary Isometric Contraction.

Studies focusing on the effects of daytime napping opportunity after partial sleep-deprived condition (n = 6) and those with no sleep-deprived condition (n = 6) are described in Section 1 and Section 2, respectively.

Section 1: nap after partial sleep-deprived condition

These studies were conducted between 2007 and 2020. Characteristics of the studies are described as follows (Table 1):

Study design

Randomized controlled trials (Waterhouse et al., 2007; Hammouda et al., 2018; Brotherton et al., 2019; Daaloul, Souissi & Davenne, 2019; Romdhani et al., 2020) with crossover design (Ajjimaporn, Ramyarangsi & Siripornpanich, 2020).

Participants

Six included studies had 67 (all male) participants. The sample size of studies ranged from nine to 15. Subjects of four studies were non-habitual nappers (Hammouda et al., 2018; Brotherton et al., 2019; Romdhani et al., 2020; Ajjimaporn, Ramyarangsi & Siripornpanich, 2020). One study included habitual nappers (Daaloul, Souissi & Davenne, 2019), and the regular napping status was not specified in another research (Waterhouse et al., 2007). All of these studies worked only on male subjects, and hence, this serves as a standard limitation. Except for one study (Ajjimaporn, Ramyarangsi & Siripornpanich, 2020), the remaining five studies lack sample size and power evaluation data.

Interventions

Duration of sleep deprivation varied between 3 to 4 h. The sleep timing ranged from 22:00–03:30 h, with one study reporting sleep timing between 03:30–06:30 h (Brotherton et al., 2019). Duration of nap ranged from 20 to 90 min, with nap timings varying between 13:00–14:30 h. Test timings ranged from 14:00 to 17:00 h.

All six studies assessed the results of different conditions amid nap and no-nap activity after sleep deprivation. At the same time, three studies evaluate the results with normal night sleep conditions (Brotherton et al., 2019; Romdhani et al., 2020; Ajjimaporn, Ramyarangsi & Siripornpanich, 2020).

Outcome measures

The diverse outcomes of sports performance measured in the studies included handgrip strength (Waterhouse et al., 2007; Brotherton et al., 2019) components of RAST, i.e., running-based anaerobic sprint test (Hammouda et al., 2018; Romdhani et al., 2020; Ajjimaporn, Ramyarangsi & Siripornpanich, 2020), sprint time (2-m, 20-m) (Waterhouse et al., 2007), single-leg jump (SJ), counter-movement jump (CMJ) and time-to-exhaustion (TTE) during karate specific test (KST) (Daaloul, Souissi & Davenne, 2019), bench press, leg press (Brotherton et al., 2019) and leg strength (Ajjimaporn, Ramyarangsi & Siripornpanich, 2020).

Quality assessment

Of the six studies, two scored 6/10 (Brotherton et al., 2019; Ajjimaporn, Ramyarangsi & Siripornpanich, 2020) and the remaining four scored 5/10 (Romdhani et al., 2020; Waterhouse et al., 2007; Daaloul, Souissi & Davenne, 2019). None of the studies fulfilled the criteria of allocation concealment, subjects, therapist, and assessor blinding. Two studies reported the dropout rates >15% (Romdhani et al., 2020; Hammouda et al., 2018). Three studies met the point and viability measures standard (Brotherton et al., 2019; Ajjimaporn, Ramyarangsi & Siripornpanich, 2020; Romdhani et al., 2020) (Table 3).

Table 3 Quality assessment of individual studies.

	
Reference	Random allocation	Concealed allocation	Baseline similarity	Subjects blinding	Therapist blinding	Assessor blinding	<15% dropouts	Intention to treat	Between-group differences	Point measures and measures of variability	Total Score	Quality rating	
													
Waterhouse et al. (2007)	1	0	1	0	0	0	1	1	1	0	5/10	Fair	
Hammouda et al. (2018)	1	0	1	0	0	0	0	1	1	1	5/10	Fair	
Daaloul, Souissi & Davenne (2019)	1	0	1	0	0	0	1	1	1	0	5/10	Fair	
Brotherton et al. (2019)	1	0	1	0	0	0	1	1	1	1	6/10	Good	
Romdhani et al. (2020)	1	0	1	0	0	0	0	1	1	1	5/10	Fair	
Ajjimaporn, Ramyarangsi & Siripornpanich (2020)	1	0	1	0	0	0	1	1	1	1	6/10	Good	
Petit et al. (2014)	1	0	1	0	0	0	0	1	1	0	4/10	Fair	
Blanchfield et al. (2018)	1	0	1	0	0	0	1	1	1	0	5/10	Fair	
Boukhris et al. (2019)	1	0	1	0	0	0	1	1	1	1	6/10	Good	
Abdessalem et al. (2019)	1	0	1	0	0	0	1	1	1	1	6/10	Good	
Hsouna et al. (2019)	1	0	1	0	0	0	1	1	1	1	6/10	Good	
Boukhris et al. (2020)	1	0	1	0	0	0	1	1	1	1	6/10	Good	

Effect of nap on sports performance

Naps significantly improved 2 and 20-m sprint times (Waterhouse et al., 2007) post-KST SJ and CMJ along with TTE during KST (Daaloul, Souissi & Davenne, 2019) components of both, bench press with average power and peak velocity along with leg press with average power (Brotherton et al., 2019) and leg muscle strength (Ajjimaporn, Ramyarangsi & Siripornpanich, 2020). Components of RAST—minimal and moderate/mean power (Ajjimaporn, Ramyarangsi & Siripornpanich, 2020), maximal power (Hammouda et al., 2018) and fatigue index (Romdhani et al., 2020) also improved with naps. However, one study reported no improvement in fatigue index post-nap (Ajjimaporn, Ramyarangsi & Siripornpanich, 2020). In addition, naps were found to have no significant effect on handgrip strength although a study contradicts this finding (Waterhouse et al., 2007; Brotherton et al., 2019) (Table 1).

Section 2: nap after no sleep-deprived condition

These studies were conducted between 2014 and 2020. Study characteristics are described as follow (Table 2):

Study design

Randomized controlled trials (Petit et al., 2014; Abdessalem et al., 2019; Hsouna et al., 2019; Boukhris et al., 2019; Boukhris et al., 2020) with crossover design (Boukhris et al., 2020; Blanchfield et al., 2018).

Participants

There were 96 participants in six included studies. The number of participants in the studies ranged from 11 to 20 digits. The gender of the subject was not specified in one study (Boukhris et al., 2020) but the rest of the studies involved only male participants, making that a common limiting point. Only one study (Petit et al., 2014) specified the status of habitual napping of participants. Four out of six studies incorporated data on sample size and power evaluation (Boukhris et al., 2020; Blanchfield et al., 2018; Boukhris et al., 2019; Hsouna et al., 2019).

Interventions

Duration of nap ranged from 20 to 90 min, and the timing of nap ranged between 13:00–16:50 h; one study reported nap time between 08:00–09:00 h in advanced phase condition (Petit et al., 2014). Test timing ranged from 15:30–17:00 h; Petit et al. (2014) reported test timing between 10:30–12:30 h in phase-advanced conditions.

All the studies, except one (Petit et al., 2014), compared the results between nap and no-nap conditions as well as between different nap durations. In addition, Petit et al. (2014) reported results on nap effectiveness after a normal night sleep and phase-advanced conditions.

Outcome measures

A variety of sports-related outcome measures were evaluated in these studies, including Wingate test (Petit et al., 2014), running time to exhaustion (TTE) (Blanchfield et al., 2018) 5-jump test (mean stride) (Hsouna et al., 2019) and maximum voluntary isometric contraction (MVIC) (Boukhris et al., 2020). Two studies (Boukhris et al., 2019; Boukhris et al., 2020) assessed all the three components of 5-m shuttle test run, i.e., total distance (TD), fatigue index (FI), best or highest distance (BD or HD), while one study evaluated only two of its components, i.e., HD and TD (Abdessalem et al., 2019).

Quality assessment

Four out of six studies scored 6/10 (Abdessalem et al., 2019; Hsouna et al., 2019; Boukhris et al., 2019; Boukhris et al., 2020), one study scored 5/10 (Blanchfield et al., 2018) and the other scored 4/10 (Petit et al., 2014). None of the six studies met the criteria of allocation concealment, subjects, therapist, and assessor blinding. One study reported dropout rates >15% (Petit et al., 2014). The criterion of point measures and measures of variability was not fulfilled by two studies (Petit et al., 2014; Blanchfield et al., 2018) (Table 3).

Effect of nap on sports performance

Components of sports performance were significantly improved in most of the studies that were included. Naps showed significant improvement in running TTE (Blanchfield et al., 2018), mean stride in the 5-jump test (Hsouna et al., 2019), MVIC (Boukhris et al., 2020), and BD/HD, TD (Abdessalem et al., 2019; Boukhris et al., 2019; Boukhris et al., 2020) along with FI (Boukhris et al., 2019; Boukhris et al., 2020) component of 5-m shuttle test run. However, Petit and colleagues reported an insignificant Effect of nap on components of the Wingate test in both normal sleep and phase-advanced conditions (Petit et al., 2014) (Table 1).

Discussion

This systematic review of the literature provides extensive insights and data related to the characteristics, outcomes and quality of clinical findings of evaluating the effects of daytime napping on components of sports performance in physically active individuals. However, the direct pooled analysis was restricted in the present review due to heterogeneity in the outcome variables. The results from the existing body of evidence suggest that daytime napping had a favourable effect on sports performance in physically active individuals, as indicated by various sports-related variables.

Effect of daytime napping on sports performance

In the present assessment, most of the studies reported favourable results of daytime napping on sports performance (Waterhouse et al., 2007; Hammouda et al., 2018; Daaloul, Souissi & Davenne, 2019; Brotherton et al., 2019; Romdhani et al., 2020; Ajjimaporn, Ramyarangsi & Siripornpanich, 2020; Blanchfield et al., 2018; Abdessalem et al., 2019; Boukhris et al., 2019; Boukhris et al., 2020; Hsouna et al., 2019), while one study (Petit et al., 2014) revealed the insignificant effect of napping, possibly due to a different methodological approach and shorter nap duration.

The appropriate degree of sleep is identified as a crucial component for athletic preparation with its importance in achieving adequate recovery, and hence optimizing athletic performance (Halson, 2008; Leeder et al., 2012). Nap is considered a recovery or revitalization period. The tendency to sleep in response to the post-lunch period with reduced core temperature and vigilance is associated with it (Chtourou et al., 2019). Therefore, napping can be considered a self-administered and cost-effective recovery method that helps to improve sports performance with bare minimum adverse effects.

Underlying mechanisms

The underlying mechanisms of how daytime napping enhances sports performance are not yet clear. However, various researchers have attempted to describe the experimental findings and clinical relevance of these napping strategies (Fig. 2).

Figure 2 The underlying mechanisms of how daytime napping enhances sports performance.

Napping opportunities, as demonstrated in studies, reduced the sense of effort, i.e., Rating of Perceived Exertion (RPE) (Blanchfield et al., 2018) and fatigue levels (Boukhris et al., 2020) and thus imparted positive effects on components of sports performance—TTE exercise and 5 m-shuttle run test. Additionally, napping influenced leg strength with decreased sleepiness perception and fatigue and improved attention (Ajjimaporn, Ramyarangsi & Siripornpanich, 2020). The non-affirmative relationship between fatigue and performance variables, as reported by Boukhris and colleagues, supported these findings (Boukhris et al., 2020). In conjunction with this, a study revealed that napping strategies decrease discrepancies in performance instigated by KST in SJ and CMJ, imparting ergogenic and psychogenic effects in managing fatigue in the athletic population (Daaloul, Souissi & Davenne, 2019).

Various studies have reported that slow-wave sleep plays a significant role in improving sports performance after napping. Napping strategies were found to be significantly beneficial in improving the indicators of RAST as well as the 5 m-shuttle run test (Hammouda et al., 2018; Romdhani et al., 2020; Ajjimaporn, Ramyarangsi & Siripornpanich, 2020). The possible credits were given to the metabolic recovery associated with longer nap durations having a more significant slow-wave component of sleep (Hammouda et al., 2018; Romdhani et al., 2020; Abdessalem et al., 2019; Boukhris et al., 2019; Boukhris et al., 2020). However, the results of the two findings have distinctions (Hammouda et al., 2018; Ajjimaporn, Ramyarangsi & Siripornpanich, 2020), which may be due to variations in nap timing and duration and the time since nap. Another study reported significant improvement in the mean stride of the 5-jump test after naps of different durations, revealing longer nap durations to have more slow-wave sleep. Thus, impart more significant benefits on attention, physical performance, and sleepiness (Hsouna et al., 2019; Lovato & Lack, 2010).

Brotherton et al. (2019), in their extensive study, reported favourable effects of nap on components of bench press and leg press, through improvement in sleepiness, alertness and tiredness. However, variables of bench press were more affected than those of leg press, as a result of effective sleep loss being complex in lifts with higher skill-orientation and with a more significant cognitive component, i.e., bench press in this study (Brotherton et al., 2019; Drust et al., 2005; Reilly & Piercy, 1994). Napping had equivocal effects on grip strength as reported by two studies (Brotherton et al., 2019; Waterhouse et al., 2007), with the inconsistent finding estimated due to differences in nap duration. Also, sprint times were found be to be improved with napping (Waterhouse et al., 2007). Nevertheless, another study concluded that there are no significant impacts of nap on performance in Wingate test, giving possible account to shorter nap duration (Petit et al., 2014).

Daytime napping in conditions with and without partial-sleep deprivation

Sleep deprivation causes a decrease in the evening rise of outcomes of athletic performance (Mah et al., 2011). While sleep extension strategies optimize the performance variables (Souissi et al., 2003). In the current review, it is reported that daytime napping improves performance in conditions with and without partial-sleep deprivation. However, it is important to consider the duration of nap, time since nap and the occurrence of sleep inertia for optimum recovery and performance enhancement.

Methodological limitations

The majority of clinical trials included in the current review demonstrated favourable improvements in sports performance. However, these studies have various essential limitations in their methodologies. Out of twelve studies, six had an average fair quality (Blanchfield et al., 2018; Petit et al., 2014; Waterhouse et al., 2007; Hammouda et al., 2018; Romdhani et al., 2020; Daaloul, Souissi & Davenne, 2019). All the studies lacked allocation concealment and subjects, therapist, and assessor blinding. These are crucial limitations, increasing the risk of bias in clinical trials. Majority of studies conducted have only male subjects, which may impact the generalizability of results on the female population. The sample size of studies was small, with only five trials reporting data on the sample size and power evaluation (Blanchfield et al., 2018; Boukhris et al., 2019; Boukhris et al., 2020; Hsouna et al., 2019; Ajjimaporn, Ramyarangsi & Siripornpanich, 2020). Chronotype of subjects, which is critical to take into account as it can affect the study outcomes, was considered only in five studies (Petit et al., 2014; Daaloul, Souissi & Davenne, 2019; Blanchfield et al., 2018; Brotherton et al., 2019; Romdhani et al., 2020).

Strengths and limitations

The article reviewed literature related to exclusive studies on the impacts of daytime napping on sports performance in healthy individuals. The review has provided information on studies incorporating both partially sleep-deprived and well-slept subjects, widening the range of the target population. However, it is crucial to emphasize appropriate sample size, allocation concealment, blinding to minimize the effect of cofounders and risk of bias on dependent variables of clinical studies.

Implications and future recommendations

The study and investigation through the clinical trials on the effects of daytime napping on sports performance should be conducted with larger sample size, considering subjects’ chronotype and travel history, with an objective assessment of prior sleep status. Therefore, it is also essential to focus on sample size and power evaluation in future researches. In addition, it is crucial to include female participants to draw valuable inferences regarding female subjects, gender differences (if any) with a clearer picture of the generalizability of results. Also, future studies should give importance to investigating the best nap duration and timing, time since nap, and test timing which cause improvements in sports-related variables and optimize aerobic and anaerobic performances.

Conclusion

The present systematic review concluded that napping strategies during the daytime improve sports performance in physically active individuals. Furthermore, napping imparts critical benefits in both partial sleep-deprived and well-slept individuals, with longer naps (~90 min) offering more significant advantages. Again, daytime naps seem to serve as an economical, easy-to-implement recovery strategy with bare minimum adverse effects in physically active, athletic population. It is, therefore, critical for the relevant stakeholders and policymakers, coaches, sportspersons, and athletes to reflect upon daytime napping as a recovery method, which when incorporated with the routine practise and training sessions, aids in reducing the accumulative effect of training-induced fatigue and thus, optimizes the sports performance as a whole.

Key points

Sleep is body’s natural recovery process where metabolic and other physiological processes slow down, aiding in revitalization of body functions and prepare the body for upcoming physiological demands. In athletic population, it is considered as the single best method of recovery to-date.

Sleep deprivation, either partial or complete, has negative effects on outcomes of sports performance. Hence, various sleep interventions have been designed to improve poor sleep patterns and optimize recovery and performance measures.

Napping is the period of <50% of night sleep duration with a tendency to fall asleep in response to the post-lunch dip of human circadian rhythm. Daytime napping improves the outcomes of sport performance in individuals with and without partial-sleep deprivation with longer nap durations (~90 min) imparting significant benefits.

The proposed underlying mechanisms include the reduction in the Rating of Perceived Exertion (RPE) and perception of sleepiness, and metabolic recovery of body functions associated with the slow—wave component of sleep. However, future studies are required to draw further valuable observations and inferences.

It is critical for the relevant stakeholders and policymakers, coaches, sportspersons, and athletes to reflect upon daytime napping as a period of revitalisation, which when incorporated with the routine practise and training sessions, aids in reducing the accumulative effect of training-induced fatigue and thus, optimizes the sports performance as a whole.

Supplemental Information

Supplemental Information 1 PRISMA checklist.

Click here for additional data file.

Additional Information and Declarations

Competing Interests

Author Contributions

Data Availability

Shibili Nuhmani is an Academic Editor for PeerJ.

Priya Sirohi conceived and designed the experiments, performed the experiments, prepared figures and/or tables, authored or reviewed drafts of the article, and approved the final draft.

Moazzam Hussain Khan conceived and designed the experiments, performed the experiments, prepared figures and/or tables, authored or reviewed drafts of the article, and approved the final draft.

Saurabh Sharma conceived and designed the experiments, authored or reviewed drafts of the article, and approved the final draft.

Shibili Nuhmani analyzed the data, prepared figures and/or tables, and approved the final draft.

Wafa Hashem Al Muslem analyzed the data, authored or reviewed drafts of the article, and approved the final draft.

Turki Abualait analyzed the data, authored or reviewed drafts of the article, and approved the final draft.

The following information was supplied regarding data availability:

The study is a systematic review without raw data.

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
