# Peer review of "A systematic review of effects of daytime napping strategies on sports performance in physically active individuals with and without partial-sleep deprivation"

_PeerJ, doi:10.7717/peerj.14460_

## Round 0.1 · original submission · Major Revisions

Thank you for submitting the manuscript to PeerJ. It has been reviewed by experts in the field and we request that you make major revisions before it is processed further.

It is recommended not to cite the articles proposed by the reviewers, one of the evaluators proposed several articles of his own, please ignore these citations. Thank you for understanding.

We look forward to hearing from you soon.

Best wishes,

Badicu Georgian, Ph.D

·

Basic reporting

No coment

Experimental design

No comment

Validity of the findings

No comment

Additional comments

Revision of the article « Effects of Daytime Napping Strategies on Sports Performance
in Physically Active Individuals with and without Partial-Sleep
Deprivation: A Systematic Review».
The article aimed to review and study the impacts of daytime napping on various variables of
sports performance in physically active individuals with and without partial-sleep deprivation
Thus, new findings in this regard may be important for scientists, sports scientist, fitness coaches, physicians, and sleep scientists.
The manuscript presents interesting findings. However, the manuscript presents some lacunes and need some improvement before acceptance.
This article fits the scope of PeerJ, but a minor revision is required before acceptance.
1. Abstract: Lane 40 "to review and study" please choose "review" or "access" or "study". The same remark applies for the end of introduction (Lane 109) in presenting the aim of your study.
2. Lane 43: Methods: replace "an electronic search" by "a systematic search".
3. Some keywords are duplicated with the title, so please replace them.
I suggest: sleep restriction, alertness, psycho-stimulants, midday sleep, cognitive performance…
4. Introduction: the introduction is missing some up-to-date references.
Kindly read those manuscripts mentioned below and add them in the introduction and discussion.

Romdhani, M., Dergaa, I., Moussa-Chamari, I., Souissi, N., Chaabouni, Y., Mahdouani, K., ... & Hammouda, O. (2021). The effect of post-lunch napping on mood, reaction time, and antioxidant defense during repeated sprint exercice.Biology of Sport, 38(4), 629.
Romdhani, M., Souissi, N., Dergaa, I., Moussa-Chamari, I., Abene, O., Chtourou, H., ... & Hammouda, O. (2021). The effect of experimental recuperative and appetitive post-lunch nap opportunities, with or without caffeine, on mood and reaction time in highly trained athletes. Frontiers in Psychology, 3955.
Souabni, M., Hammouda, O., Romdhani, M., Trabelsi, K., Ammar, A., and Driss, T. (2021). Beneits of daytime napping opportunity on physical and cognitive performances in physically active participants: a systematic review. Sport. Med. In press. 15, 874–883. doi: 10.1007/s40279-021-01482-1
5. Introduction: Lane 99: "Sleep loss, fatigue, and stress can increase the measure of sleepiness…" Please update the reference (Winget, DeRoshia & Holley, 1985), you may use one of the requested one above.
6. Lane: 341 Implications and Future Recommendations:
A. You may mention that future studies may investigate the impact of daytime napping strategies during different lunar phases on sports performance since it has been reported recently that the lunar cycle deeply impacts sleep (you may use the references below).
Dergaa, I., Fessi, M. S., Chaabane, M., Souissi, N., & Hammouda, O. (2019). The effects of lunar cycle on the diurnal variations of short-term maximal performance, mood state, and perceived exertion. Chronobiology international, 36(9), 1249-1257.
Dergaa, I., Romdhani, M., Fessi, M. S., Ben Saad, H., Varma, A., Ben Salem, A., ... & Hammouda, O. (2021). Does lunar cycle affect biological parameters in young healthy men?. Chronobiology International, 38(6), 933-940.
Benedict, C., Franklin, K. A., Bukhari, S., Ljunggren, M., & Lindberg, E. (2022). Sex-specific association of the lunar cycle with sleep. Science of the Total Environment, 804, 150222.
Hartstein, L. E., Wright Jr, K. P., Akacem, L. D., Diniz Behn, C., & LeBourgeois, M. K. (2022). Evidence of circalunar rhythmicity in young children's evening melatonin levels. Journal of Sleep Research, e13635.
B. You may also mention that future studies should investigate the impact of a combination of daytime napping strategies and daytime melatonin supplementation to curtail the impact of sleep deprivation (reference below).
Souissi, A., Dergaa, I., Chtourou, H., & Ben Saad, H. (2022). The Effect of Daytime Ingestion of Melatonin on Thyroid Hormones Responses to Acute Submaximal Exercise in Healthy Active Males: A Pilot Study. American Journal of Men's Health, 16(1), 15579883211070383.
Souissi, A., Dergaa, I., Musa, S., Saad, H. B., & Souissi, N. (2022). Effects of daytime ingestion of melatonin on heart rate response during prolonged exercise. Movement & Sport Sciences-Science & Motricité, (115), 25-32.
7. The manuscript needs to be reviewed by a fluent English speaker.

·

Basic reporting

First of all, I would like to congratulate the authors for their efforts.

There are some areas in this manuscript that need to be revisions.

Major:

-More detailed physiological background should be mentioned in the introduction.

-The discussion section is very weak. The discussion section needs to be completely reorganized.

-In the conclusion, messages should be given to practical practitioners to take home.

Minor:

-Table 1 should be redesigned. Difficult for readers to understand.

Best Regards.

Experimental design

It has a good design.

Validity of the findings

Yes

·

Basic reporting

It is more logical that the title goes in the reverse order, because the essence of the work is that it is a systematic review of works, so I suggest that the title reads A Systematic Review of Effects of Daytime Napping Strategies on Sports Performance in Physically Active Individuals with and without Partial-Sleep Deprivation

Experimental design

Meta analysis is correct.

Validity of the findings

No coments.

Additional comments

General Methods is Mata Analysis, "An electronic search" is only technic.

·

Basic reporting

In my opinion the title of the article should be restructured.
The introduction needs more detail with scientific information taken from the studied literature and articles.

Experimental design

I suggest the statistical interpretation of the data presented in the table no. 1.

Validity of the findings

No comment.

Additional comments

I suggest reformulation of certain phrases that are repeated in the manuscript.

---

## Round 0.2 · Major Revisions

Please revise the article according to the reports received by the reviewers.

Best regards,

·

Basic reporting

Revision of the article: "A Systematic Review of the Effects of Daytime Napping Strategies on Sports Performance in Physically Active Individuals with and without Partial-Sleep Deprivation".

The purpose of this study was to investigate the impact of daytime napping on various variables of sports performance in physically active people with and without partial sleep deprivation. 

This article falls within the scope of PeerJ, but it must undergo a thorough review before being accepted.


Before I start going through the manuscript in detail and pointing out the small mistakes, a few big concerns come to mind.

The authors submitted their manuscript in 2022, not 2020. Their choice of articles conducted between 2007 and 2020 makes no sense to me, unless the authors are too lazy to re-conduct their meta analysis and/or to adjust the content of their article accordingly.

What is the novelty of your systematic review if it does not include the most recent scientific evidence? We're almost to 2023, so there's a three-year gap in scientific research. By analysing what was done three years ago and ignoring the research from the previous three years, you will not add anything to scientific research. 

And what makes matters worse is that a lot of attention has been paid to the impact of daytime napping on  sports performance during 2021 and 2022.

Furthermore, during my previous review process, I suggested that the manuscript be revised by a fluent English speaker, which, as far as I can see, was not done.

Experimental design

Article content is within the Aims and Scope of the journal. But, the investigation wasn't conducted rigorously.

Validity of the findings

There is no novelty of the study since the systemic review didn't take into consideration the three previous years.

Additional comments

No additional comments.

·

Basic reporting

The authors did all revisions adequately.

The article can be accepted if it is suitable for other reviewers and the editor.

Experimental design

The authors did all revisions adequately.

Validity of the findings

Good

---

## Round 0.3 · accepted · Accept

Accepted for publication. Congratulations!